# Targeting Iron-Sulfur Clusters in Cancer: Opportunities and Challenges for Ferroptosis-Based Therapy

**DOI:** 10.3390/cancers15102694

**Published:** 2023-05-10

**Authors:** Jaewang Lee, Jong-Lyel Roh

**Affiliations:** Department of Otorhinolaryngology-Head and Neck Surgery, CHA Bundang Medical Center, CHA University, Seongnam 13488, Republic of Korea

**Keywords:** iron, ferroptosis, iron-sulfur cluster, cancer, therapy

## Abstract

**Simple Summary:**

Iron-sulfur clusters (ISCs) play a crucial role in cancer cell survival and growth, and their dysregulation is associated with the development and progression of cancer. This review highlights the link between ISC metabolism and ferroptosis, a new form of regulated cell death induced by iron-dependent accumulation of lethal lipid peroxidation. The review summarizes the current knowledge on the mechanisms of ISC biogenesis, the role of ISC modulation in ferroptosis, and the potential of targeting ISCs for cancer therapy. The review provides insight into new therapeutic strategies for cancer treatment based on regulating ISC metabolism.

**Abstract:**

Iron dysregulation is a hallmark of cancer, characterized by an overexpression of genes involved in iron metabolism and iron-sulfur cluster (ISC) biogenesis. Dysregulated iron homeostasis increases intracellular labile iron, which may lead to the formation of excess cytotoxic radicals and make it vulnerable to various types of regulated cell death, including ferroptosis. The inhibition of ISC synthesis triggers the iron starvation response, increasing lipid peroxidation and ferroptosis in cancer cells treated with oxidative stress-inducing agents. Various methods, such as redox operations, iron chelation, and iron replacement with redox-inert metals, can destabilize or limit ISC formation and function, providing potential therapeutic strategies for cancer treatment. Targeting ISCs to induce ferroptosis represents a promising approach in cancer therapy. This review summarizes the state-of-the-art overview of iron metabolism and ferroptosis in cancer cells, the role of ISC modulation in ferroptosis, and the potential of targeting ISCs for ferroptosis induction in cancer therapy. Further research is necessary to develop and validate these strategies in clinical trials for various cancers, which may ultimately lead to the development of novel and effective treatments for cancer patients.

## 1. Introduction

Ferroptosis is a form of regulated cell death (RCD) characterized by the accumulation of lipid peroxides, which are formed through the oxidation of polyunsaturated fatty acids in cell membranes [1]. The accumulation of lipid peroxides can lead to membrane damage and cell death. This form of RCD was first reported in 2012 and was subsequently recognized as a distinct mechanism of cell death in 2018 [2,3]. Since then, ferroptosis has gained increasing attention in the scientific community due to its unique morphological and biochemical features that differentiate it from other forms of RCD. Ferroptosis is initiated by intracellular perturbations in iron metabolism and redox homeostasis. The accumulation of labile iron and the subsequent generation of reactive oxygen species (ROS) through the Fenton reaction trigger oxidative damage of membrane lipids, particularly polyunsaturated fatty acids, leading to lipid peroxidation and, ultimately, ferroptotic cell death [1]. Endogenous radical-trapping antioxidant systems play a vital role in countering the accumulation of lipid peroxidation by converting cellular ROS to H_2_O [4]. Ferroptosis is regulated by canonical regulators, such as the cystine/glutamate antiporter system xc^–^ (xCT), glutathione peroxidase 4 (GPX4), ferroptosis suppressor protein 1 (FSP1)-CoQ_10_, and dihydroorotate dehydrogenase (DHODH), which have primary antioxidant functions and can detoxify cellular lipid peroxidation [5] (Figure 1).

Iron is an essential micronutrient involved in a plethora of biological processes, including DNA synthesis, cellular respiration, and immune function [6]. Iron homeostasis is tightly regulated in healthy cells to balance systemic absorption and distribution and cellular uptake, storage, and export [7]. The dysregulation of iron homeostasis has been implicated in the development of various diseases, including cancer [8]. Recent studies have shown that cancer cells require increased intracellular iron concentrations for their growth and survival, particularly in highly malignant or stem-like cancer cells [9]. The intracellular labile iron pool (LIP) is elevated in cancer cells and is required for tumor growth and metastasis [10]. However, increased iron levels can also generate highly reactive radicals via the Fenton reaction, which can cause DNA damage and promote tumor oxidative cell death.

Iron-sulfur clusters (ISCs) are essential components that consist of iron (Fe) and inorganic sulfur (S) and participate in various biological processes [11]. To protect them from the damaging effects of high oxygen levels, ISCs have evolved in biogenesis systems [12]. ISCs play crucial roles in electron transfer, enzyme catalysis, and biological regulation [13]. Cellular iron is utilized for iron-sulfur biogenesis, heme synthesis, and iron-containing proteins [14]. Heme synthesis and DNA metabolic enzymes require interaction with ferrochelatases (FECH) and other ISCs to function in DNA polymerases and helicases [14,15]. As iron is essential for forming ISCs, cellular iron regulation and ISC modulation are crucial for inducing ferroptosis in cancer cells [16]. Therefore, targeting iron metabolism and ISC modulation may provide a promising therapeutic strategy for cancer treatment [17]. Recent studies have demonstrated that targeting ISC biogenesis can induce ferroptosis and reduce tumor growth in preclinical cancer models [18]. Therefore, the potential of targeting ISCs for cancer treatment by inducing ferroptosis needs to be highlighted. This review aims to provide an overview of iron metabolism and ferroptosis in cancer cells, the role of ISC modulation in ferroptosis, and the potential of targeting ISCs for ferroptosis induction in cancer therapy.

## 2. Iron Metabolism and Ferroptosis in Cancer

Iron metabolism is altered in cancer cells to promote tumor initiation, proliferation, and metastasis [19]. Tumor-initiating and cancer stem cells tend to accumulate more cellular iron by increasing iron absorption and storage while decreasing iron efflux [20]. This is associated with a decrease in ferroportin (FPN) and an increase in TFR1, STEAP, DMT1, and IRP2 in several types of cancers [21]. Iron is essential for many cellular processes, such as DNA synthesis and metabolism, cell cycle progression, energy production, and mitochondrial function [17]. Ribonucleotide reductase, an iron-dependent protein, is involved in DNA replication, highlighting the importance of iron in cancer growth. Iron also plays a critical role in cancer progression by contributing to pathways, such as the cell cycle, ATP production, mitochondrial oxygen consumption, and citric acid cycling [22]. Dysregulated iron homeostasis creates a favorable microenvironment for tumor growth and metastasis, making targeting iron metabolism a potential therapeutic approach in cancer treatment. Effective therapies are needed to regulate cellular iron accumulation and induce ferroptosis [16]. Further studies are necessary to develop these therapies and understand their efficacy in treating cancer.

The transportation of cytosolic labile iron is primarily directed toward the mitochondria, which is used in heme and ISC biogenesis pathways. The mitochondria are responsible for oxygen consumption and electron transfer, and labile iron in the mitochondria plays a crucial role in generating ROS through the Fenton reaction. Heme and ISC are the primary components of mitochondrial iron, while excess iron is stored in nuclear-encoded mitochondrial FT [23]. Frataxin (FXN), an iron donor protein, plays a vital role in ISC assembly and functions as an iron chaperone that regulates pro-oxidation through aconitase activity [24]. The dysregulation of mitochondrial iron homeostasis can lead to changes in mitochondrial and cellular LIP levels, potentially resulting in cardiovascular or neurodegenerative disorders.

The cellular metabolism of iron involves the regulation of cytosolic labile iron, a minor but crucial fraction of the total redox-active iron in the cell [25] (Figure 1). Iron uptake via transferrin receptor 1 (TFR1) and storage within ferritin (FT) are essential in regulating the cytosolic labile iron pool. Divalent cation transporter 1 (DMT1) transports iron to the cytosol upon its endocytosis through TFR1, which is then reduced to ferrous iron to generate cytosolic labile iron. Poly(rC)-binding protein 1 (PCBP1) and PCBP2 transport labile iron to the mitochondria or store it within FT for future use. Ferroxidase-catalyzed oxidation converts ferrous iron to ferric iron for storage within FT, preventing cellular damage from excess cytosolic labile iron [26]. The regulation of labile iron by FT overexpression reduces reactive radical generation, whereas FT downregulation can increase LIP and oxidative damage. Moreover, FT mobilizes to lysosomes to allow the iron to be recycled when iron metabolic demand is high.

During the process of autophagy, iron-containing molecules such as FT are degraded within the lysosomes, resulting in a significant amount of redox-active labile iron [27]. The acidic lysosomal environment, aided by an ATP-dependent proton pump, facilitates this process and recycles iron for biosynthesis [28]. DMT1 helps to release ferrous iron from late endosomes and lysosomes into the cytosol [29]. Active autophagy is triggered when intracellular iron is lacking, releasing iron from lysosomes through FT degradation [30]. However, the autophagy deficit pathway has a limited effect on intracellular iron levels. Lysosomes mainly contain ferrous iron that can lead to cytotoxicity if the lysosomal membrane is damaged and prone to peroxidation [31]. Deferoxamine (DFO), taken up through endocytosis, chelates lysosomal labile iron and stabilizes lysosomes against oxidative stress, thereby inhibiting DNA damage and cell death [30]. Autophagolysosomal degradation of FT depends on the nuclear receptor coactivator 4 (NCOA4), which is enriched in autophagosomes [32] and is crucial for regulating cellular LIP levels and maintaining iron homeostasis.

Ferritinophagy is a cellular process where FT is degraded through autophagy by being bound to NCOA4 within autophagosomes [33]. The lysosomal proteolysis activity induced by NCOA4 liberates iron from FT, thereby regulating the cellular iron balance by controlling the balance between reduced and oxidized ferric iron [34]. Autophagy-related (ATG) proteins co-localize with NCOA4 during autophagosome formation [33]. An abnormal iron balance caused by dysfunctional ferritinophagy results in toxic oxidative stress and various pathologies, such as neurodegenerative diseases [35]. NCOA4 plays a critical role in ferroptosis initiation [36]. Erastin, an oncogenic RAS-selective lethal chemical, is known to induce ferroptosis and was first reported in 2012 [2]. However, NCOA4-mediated ferritinophagy-induced ferroptosis differs from other RCD processes regarding morphological and biochemical characteristics [37,38]. The increased ferrous iron resulting from excessive lipid peroxidation during NCOA4-mediated ferritinophagy leads to the destruction of cell membranes and cell death [39]. The regulation of ferroptosis depends on two major antioxidant system molecules: GPX4 and xCT [1]. GPX4 is a cellular antioxidant that reduces lipid hydroperoxides, while xCT is a membrane protein that detoxifies lipid peroxidation by facilitating the intracellular uptake of cystine for GSH production. NCOA4-mediated ferritinophagy is activated in response to changes in intracellular iron levels, leading to FT degradation [40]. The absence of NCOA4 causes a drop in iron levels. A high iron concentration in the cell accelerates the degradation of NCOA4 through the ubiquitin ligase activity of HECT E3 ubiquitin ligase 2 (HERC2) and E6AP carboxyl terminus, blocking the release of excess iron and ferritinophagy. In ferritinophagy, FBXL5, an F-box/LRR-repeat protein 5, may also play a role in ferritin (FT) degradation by targeting IRP2 [41].

Iron dysregulation plays a crucial role in the cancer phenotype, particularly in the “iron addiction” phenotype [42]. Dysregulated iron homeostasis, caused by the overexpression of TFRs, a loss of FPN expression, and the abnormal expression of iron-regulating genes, leads to an increase in the concentration of intracellular LIP due to iron overload. The increased levels of LIP lead to persistent oxidative stress caused by loosely bound ferrous iron, which cancer cells protect against through various mechanisms. FT stores labile iron in ferric form and reduces cellular LIP content. However, IRP1 and IRP2 post-transcriptionally inhibit FT, which modulates the cellular labile iron content [26]. In some cancers, FT expression is upregulated, which alters the cellular labile iron content [43]. Heme oxygenase-1 (HO-1) is an enzyme that breaks down heme into biliverdin/bilirubin, carbon monoxide, and ferrous iron and provides a cytoprotective defense under different stress conditions [44]. However, the excessive activation of HO-1 increases LIP and ROS production, exceeding the buffer capacity of FT and resulting in cytotoxic effects. The NRF2/HO-1 axis regulates the LIP content in cancer cells, and HO-1 activation increases LIP and lipid peroxidation, triggering ferroptosis [45]. Mitochondrial electron transport chain complex IV, cytochrome c oxidase, reduces cellular LIP and lipid peroxidation and enhances the cytoprotective effect of cancer cells when activated [46]. The inhibition of phosphoinositide-3-kinase affects the interaction between the iron-responsive element (IRE) and the iron regulatory protein (IRP), increases the expression of FT mRNA, and leads to the depletion of LIP in cancer cells [47].

## 3. ISC Modulation and Ferroptosis in Cancer

Cancer cells have a distinct dependence on iron compared to nonmalignant cells, which may stem from their reliance on ISC biogenesis systems [48]. The dysregulation of genes involved in ISC synthesis and transport has been observed in different cancer types [48], indicating the potential significance of ISC modulation in cancer therapy, which heavily relies on RCD for success.

ISC biogenesis and heme synthesis occur on the inner mitochondrial membrane [49]. The generation of Fe-S clusters is a complex, multi-step process that involves the formation of clusters in the mitochondria, their export into the cytosol, and their subsequent integration into target proteins. Spectroscopic studies have identified the functional characteristics of ISC-containing proteins [50]. In mammals, ISC biogenesis consists of four steps: [2Fe-2S] cluster formation on the iron-sulfur assembly scaffold, the release and transportation of [2Fe-2S] clusters by chaperone proteins, the conversion of [2Fe-2S] clusters to [4Fe-4S] clusters, and the insertion of newly formed [4Fe-4S] clusters into apo-proteins [51]. The various steps require a cascade of ISC proteins, including cysteine desulfurases NFS1, ISU1, and ISU2, FXN, transfer protein monothiol glutaredoxin Grx5, chaperones, and other proteins, such as the Ind1 complex and the Bol proteins. Notably, research has indicated that regulating the biogenesis and transfer of ISCs can modulate ferroptosis in cancer [16].

Disruptions to the general pathways involved in the biogenesis of ISCs have significant cellular consequences, as demonstrated by a novel cellular system created using mutants of the core scaffold protein for ISC assembly, iron-sulfur cluster assembly enzyme (ISCU) [52]. This system induces an acute deficiency of ISCs and a loss of aconitase activity, leading to a 12-fold increase in intracellular citrate levels. ISC-deficient cells also exhibit an accumulation of fatty acid biosynthesis and lipid droplets. The metabolic reprogramming can predispose them to cellular steatosis and potentially initiate ferroptosis through the accumulation of lipid ROS induced by iron [53]. These findings highlight the unanticipated links between ISC machinery and human diseases. Modulating ISCs can cause metabolic reprogramming that is favorable for ferroptosis induction, providing potential targets for treating human cancers.

Cancer cells preferentially produce a protein called CISD2 (nutrient deprivation autophagy factor-1; NAF-1), a homodimer protein composed of metal and sulfur. CISD2 promotes rapid cell proliferation by regulating the mitochondrial iron and ROS levels [54]. Recent research has focused on the role of CISD2 in ferroptosis, a type of cell death involving the accumulation of mitochondrial iron and peroxidation [55]. In human breast cancer cells, the disruption of CISD2 function leads to the accumulation of mitochondrial iron and mitochondrial ROS, which can trigger ferroptotic cell death [56]. On the other hand, pioglitazone stabilizes the iron-sulfur cluster of CISD1, which inhibits mitochondrial iron uptake, lipid peroxidation, and, ultimately, ferroptosis [57]. This indicates that CISD1 has a role in safeguarding against mitochondrial damage during ferroptosis.

CISD2 has been detected in the endoplasmic reticulum (ER) and the membranes associated with mitochondria-ER, which are the operational sites of NCOA4, a receptor for cargo that mediates ferritinophagy and influences ferroptosis by modulating the labile iron and ISCs [31]. Several studies suggest a close correlation between CISD2 expression and resistance to ferroptosis in human cancers [58]. The increased expression of CISD2 reduces sensitivity to ferroptosis inducers. In contrast, the genetic knockdown of CISD2 or the administration of pioglitazone increases mitochondrial ferrous iron and lipid peroxidation, leading to the vulnerability of cancer cells to ferroptosis induction by sulfasalazine [59]. Mitochondrial iron levels are maintained by interacting with the NEET proteins CISD1 and CISD2 [60]. The upregulation of either CISD1 or CISD2 reduces the susceptibility of human cancers to ferroptosis inducers by limiting mitochondrial iron uptake [57]. A recent report suggests that CISD1 and CISD2 have two parallel mechanisms affecting tumorigenesis [61]. One mechanism is related to oxidative stress and ferroptosis inhibition, where CISD2 enhances the accumulation of free iron via ferritinophagy-dependent ferritin turnover. The other mechanism involves inhibiting the p62-Keap1-NRF2 pathway due to CISD2 depletion. Recent studies have shown that CISD3 depletion increases glutaminolysis and mitochondrial oxidative phosphorylation, rendering cancer cells or hepatocytes vulnerable to xCT inhibition [62,63]. These research findings demonstrate that CISD1, CISD2, and CISD3 have the potential to modulate ferroptotic cell death in cancer cells through the regulation of their ISCs.

NFS1 is a crucial biosynthetic enzyme involved in the de novo synthesis of [2Fe-2S] clusters, essential for early ISC machinery [64]. Research on NFS1 has revealed its vital role in regulating iron homeostasis and sensitivity to ferroptosis in cancer cells. The inhibition of NFS1 disrupts ISC biosynthesis, leading to iron starvation responses, intracellular cysteine transport inhibition, and ferroptosis activation [65]. The enzyme also contributes to lung adenocarcinoma survival in high-oxygen environments, maintaining iron-sulfur cofactors in cell-essential proteins. Inadequate ISC maintenance triggers the iron-starvation response and ferroptosis when combined with GSH biosynthesis inhibition. In vitro, NFS1 suppression combined with cysteine transport inhibition slows tumor growth by activating ferroptosis. Recent genome-wide synthetic lethality-screening assays have shown that targeting the CAIX-NFS1/xCT axis can regulate the vulnerability of solid hypoxic tumors [66]. NFS1 plays a crucial role in suppressing ferroptosis and preventing acidified intracellular pH via the tumor hypoxia-induced pH regulator carbonic anhydrase IX (CAIX), contributing to tumor progression and therapeutic resistance. NFS1 inhibition also enhances the sensitivity of colorectal cancer cells to oxaliplatin by triggering PANoptosis, a cell death process involving apoptosis, necroptosis, pyroptosis, and ferroptosis [67]. Oxaliplatin-based oxidative stress enhances S293 phosphorylation of NFS1 serine residues, preventing PANoptosis, while NFS1 inhibition improves platinum-based chemosensitivity. Recently, a novel small molecule called eprenetapopt (APR-246, PRIMA-1^MET^) has shown potential in triggering ferroptosis by inhibiting NFS1 and limiting cysteine desulfurase activity, thereby reducing ISC biogenesis and restricting cellular proliferation [68]. Together with dietary serine and glycine restriction, eprenetapopt synergizes to inhibit esophageal xenograft tumor growth.

FXN is a mitochondrial protein that is crucial for the biogenesis of ISCs. The early machinery responsible for this process forms a core complex that recruits ISD11, a protein that binds to NFS1, and eventually FXN [69]. Insufficient levels of FXN have been associated with a range of metabolic disturbances, including Friedreich’s ataxia, a neurodegenerative disease [70]. Interestingly, increased FXN expression in cancer cells has been linked to hypoxia-induced tumor stress via the HIF pathway [71]. Recent studies have highlighted the crucial role of FXN in regulating ferroptosis. The inhibition of FXN expression can interfere with the assembly of ISCs, leading to iron starvation stress and cellular ferroptosis by promoting lipid peroxidation [72]. This can result in severe mitochondrial morphological damage, such as the fragmentation and loss of cristae, which are the hallmarks of ferroptosis. However, blocking the signal of iron starvation, whether through genetic or pharmacological means, can completely restore resistance to ferroptosis in FXN knockdown cells and xenografts. Therefore, FXN is a critical regulator of ferroptosis and could be a promising target for enhancing antitumor activity through the induction of ferroptotic cell death.

Glutaredoxin 5 (GRX5) is crucial in transferring ISCs from mitochondria to downstream targets, such as ferrochelatase, IRP, and m-aconitase [73]. Mutations or deficiencies in GRX5 can lead to impaired heme biosynthesis and sideroblastic anemia by disrupting downstream ISC biosynthesis and maturation [74]. Moreover, GRX5 can activate the IRE-binding activity of IRPs under cytosolic iron depletion [75,76]. Recent research has shown that silencing GLRX5 can activate the iron starvation response, increasing intracellular free iron levels by increasing the binding of IRPs to IREs in therapy-resistant head and neck cancer cells, thereby predisposing them to ferroptosis [77]. Mutations in GLRX5 can also affect the production of downstream ISCs, leading to chemoresistance to cisplatin in various cancer cells [76]. Thus, inhibiting GLRX5 may induce ferroptosis in cisplatin-resistant cancer cells, presenting a promising therapeutic strategy to overcome chemoresistance by promoting ferroptosis via GLRX5 inhibition [77].

The inhibition of iron-sulfur cluster assembly 2 (ISCA2) at a late stage of mitochondrial ISC machinery can decrease HIF-1/2α levels and promote ferroptosis in clear cell renal cell carcinoma (ccRCC) [78]. ISCA2 inhibition by pharmacological intervention or siRNA disrupts IRE-dependent translation. It reduces HIF-2α protein levels, which activates an iron starvation response, resulting in iron/metal overload and eventual cell death through ferroptosis. Notably, pharmacological inhibitor #25 of ISCA2 significantly reduces the growth of ccRCC xenografts in vivo, decreases the HIF-α level, and enhances lipid peroxidation, indicating increased ferroptosis in vivo. Therefore, targeting ISCA2 may be a promising therapeutic approach to inhibit HIF-1/2α and promote ferroptosis in von Hippel–Lindau-deficient cancer cells.

Iron depletion from mitochondria, either through iron chelators or the knockdown of core components of the ISC assembly machinery, triggers FUNDC1-dependent mitophagy [79,80]. One of the cellular iron sensors that help maintain iron homeostasis is IRP1, which plays a critical role in mitophagy induced by iron stress [79]. Blocking the biosynthesis of ISCs can result in iron overload and facilitate cell death through ferroptosis. IRP1 and IRP2 control intracellular iron levels at the posttranscriptional level, thereby maintaining iron homeostasis and preventing the generation of ROS caused by excess iron. The suppression of ISC synthesis can activate IRP2, which may enhance the binding of IRP2 to target mRNAs independently of IRP1 and FBXL5, thereby promoting ferroptosis sensitivity [80]. Additionally, inhibiting mitochondrial ISC protein synthesis and mitochondrial respiration by suppressing iron utilization sensitizes pancreatic ductal adenocarcinoma (PDAC) cells to the MEK inhibitor [81]. The selective autophagy adaptor NCOA4 targets the cellular iron storage complex ferritin (ferritinophagy) for lysosomal degradation, thereby releasing iron for cellular utilization. Conversely, enhanced NCOA4-mediated ferritinophagy accelerates PDAC tumorigenesis by maintaining iron homeostasis in the mitochondria and increasing iron availability for ISC biogenesis [82]. The results suggest that manipulating ISC biosynthesis could be a promising approach to regulating ferroptosis sensitivity.

ISCs are a crucial group of molecules involved in maintaining mitochondrial function and regulating iron metabolism and have recently become a focus of research on cancer cell death, including ferroptosis [16]. The inhibition of ISC synthesis triggers the iron starvation response, increasing lipid peroxidation and ferroptosis in cancer cells treated with oxidative stress-inducing agents. These studies have revealed that several molecules involved in mitochondrial ISC biosynthesis serve as novel regulators of ferroptosis and may represent promising targets for enhancing antitumor activity through ferroptotic cell death.

## 4. Implications of Targeting ISCs for Ferroptosis-Based Cancer Therapy

The modulation of ISCs using pharmacological agents to trigger ferroptosis is a promising research area. Targeting ISC-related genes has been shown to activate the ferroptotic cascades, suggesting that suppressing ISC biosynthesis may induce ferroptosis in cancer cells [18]. The overexpression of several genes involved in ISC synthesis and transport has been observed in multiple types of cancer, with most [2Fe-2S] synthesis being upregulated, except for ISCU [48]. Given the critical role of ISCs in cancer metabolism, there is growing interest in using drugs that target the iron-sulfur metabolic network to disrupt cancer cell metabolism selectively and inhibit cancer progression [83,84]. Various methods can be used to destabilize ISCs, including redox operations, increasing iron chelation, and replacing iron with redox-inert metals (Figure 2). These strategies can potentially limit the function of Fe-S-containing proteins and prevent cluster formation, providing new opportunities to target ISCs during cancer initiation and progression. Taken together, targeting ISCs to induce ferroptosis represents a promising therapeutic approach for treating human cancers.

Iron chelation therapy has been a topic of interest in cancer treatment. DFO is a commonly used iron chelator with a strong affinity for Fe^3+^ and can reduce systemic iron overload [85]. Iron depletion from mitochondria leads to mitophagy and mitochondrial dysfunction, suppressing tumor growth and spread [86]. When targeted to the mitochondria (as mitoDFO) without altering systemic iron metabolism, it disrupts the biogenesis of ISCs and heme, destabilizing and reducing the activity of enzymes that require these cofactors. MitoDFO also inhibits mitochondrial respiration, ROS production, mitochondrial network fragmentation, and mitophagy induction. DFO was the first iron chelator to be tested as an anticancer agent. Studies have shown that DFO has anti-proliferative effects on leukemia cells in preclinical settings, both in vitro and in vivo [87]. However, clinical trials in neuroblastoma and prostate cancer patients did not yield significant benefits from DFO administration, likely due to poor bioavailability and subcutaneous administration [88,89]. Consequently, researchers have sought out more promising iron chelators, with deferasirox (DFX) emerging as a potential candidate. DFX is a synthetic iron chelator that forms a tridentate complex with both Fe^2+^ and Fe^3+^ and has an extended elimination half-life compared to DFO [90,91]. It is orally available, which has increased clinical interest. However, the selective depletion of mitochondrial iron in cancer cells without affecting systemic iron metabolism may also be necessary to minimize potential adverse effects in vivo, as ISC biogenesis occurs in the mitochondria. Collectively, these findings suggest that iron chelation may be worth exploring as a method of targeting Fe-S metabolism to enhance cancer therapies.

There are multiple ways in which the Fe-S metabolic network can be disrupted, including through redox manipulations that compromise the stability of ISCs. Proteins containing either a [2Fe-2S] or [4Fe-4S] cluster are particularly susceptible to these manipulations. The [4Fe-4S]^2+^ cluster in aconitase is particularly interesting due to its crucial role in the citric acid cycle [92]. The solvent-exposed Fe site of this cluster acts as a Lewis acid during the conversion of citrate to isocitrate. However, increased radicals can oxidize the [4Fe-4S]^2+^ cluster, rendering aconitase inactive and releasing a redox-active Fe atom, which can promote further oxidation reactions and lead to additional cellular damage. As such, aconitase activity is a valuable marker of intracellular ROS stress [93]. Ascorbic acid, also known as vitamin C (vitC), has shown promise as a chemical therapy for altering the stability of ISCs within cells [94]. VitC can decrease aconitase activity and act as an antioxidant due to its ability to work as a one-electron reductant [95]. However, it can also act as a pro-oxidant via autoxidation by reacting with catalytically active iron to selectively kill cancer cells [96]. While vitC can reduce Fe^3+^ to Fe^2+^, it can also generate high levels of intracellular H_2_O_2_. Clinical studies have shown that high doses of vitC, combined with standard cancer treatments, show promise as an anticancer therapy, with minimal added toxicity reported in early-phase clinical trials involving patients with various types of cancer [97,98]. In anaplastic thyroid cancer, vitC treatment has been found to induce ferroptosis by promoting GPX4 inactivation and ferritinophagy activation, leading to the release of free iron [99]. This positive feedback process mediated by ROS and iron sustains lipid peroxidation, ultimately resulting in ferroptosis and suggesting that vitC could be a viable cancer therapy option. However, caution is warranted when using vitC in conjunction with ferroptosis-based cancer therapies due to its potential anti-ferroptotic effects, similar to those of DFO and DFX [100]. Treatments that generate H_2_O_2_ and other ROS can efficiently disrupt Fe-S metabolism downstream of the ISC biogenesis process and provide a potential strategy for cancer therapy. Selectively targeting ISCs with chemical agents such as vitC could help develop new anticancer therapies.

Substituting iron with metals that do not participate in redox reactions prevents the formation and function of Fe-S-containing proteins and ISCs. Gallium (Ga) is a promising therapeutic approach that targets iron metabolism. Ga is a metal similar in size and valence structure to iron, allowing it to coordinate ligands similarly to Fe^3+^ [101]. Ga^3+^ can compete with iron for binding to FT and act as an antagonist to divalent metal ions [102]. This mechanism inhibits the function of iron-containing proteins, including ribonucleotide reductase, which may have anticancer effects [103]. Clinical trials have demonstrated the safety and efficacy of Ga^3+^-nitrate in treating advanced non-Hodgkin’s lymphoma [104]. A new derivative of gallium, called gallium maltolate (GaM), has greater bioavailability and a longer half-life, which increases its cytotoxicity against lymphoma cells [105]. GaM can impair mitochondrial function and ribonucleotide reductase activity in cancer cells, which also require cytoplasmic Fe-S protein assembly machinery for biogenesis [106,107]. Therefore, Ga^3+^ can function as an iron mimic to inhibit Fe-S biogenesis and insertion into apo-proteins.

ISC-targeting compounds have shown promise in promoting ferroptosis in cancer cells. Cysteine, which generates persulfides that are intermediates in ISC production, is an essential substrate for the cellular thiophane-sulfur production system. Plant thiane sulfide donors, such as diallyl trisulfide and dimethyl trisulfide, increase cellular resistance to ferroptosis in osteosarcoma cells treated with erastin [108]. Dihydroartemisinin induces ferritinophagy and lipid peroxidation accumulation, promoting ferroptosis activation in leukemia cells by interfering with the function of ISCU [109]. Pioglitazone, an FDA-approved drug for type 2 diabetes, stabilizes the ISC of CISD1, inhibiting mitochondrial iron uptake, lipid peroxidation, and subsequent ferroptosis [57]. Pioglitazone also sensitizes head and neck cancer cells to sulfasalazine-induced ferroptotic cell death in vitro and in vivo by inhibiting CISD2 activity [59]. Other ISC assembly pharmaceutical inhibitors, eprenetapopt or compound #25, reduce ISC biogenesis, restrict cancer cell proliferation, and trigger ferroptosis by inhibiting NFS1 or ISCA2 [68,78]. Most studies on ISC inhibitors have been involved only in the preclinical stage but have not reached clinical trials. Further validation of these efficacies for ferroptosis-based cancer therapy is needed in clinical trials. Taken together, targeting the molecules involved in ISC biogenesis and transfer could promote ferroptosis and hold promise as a strategy for treating human cancers. Therefore, further research and development of compounds targeting these mechanisms of action are urgently needed for cancer therapy.

## 5. Conclusions and Perspectives

This article comprehensively reviews iron metabolism, ferroptosis, and the potential of targeting ISCs for cancer therapy. The dysregulation of iron homeostasis is a hallmark of cancer, leading to excess reactive labile iron that can induce ferroptosis, a regulated form of cell death. While significant progress has been made in understanding the mechanisms of ferroptosis, the contribution of iron-mediated ROS production and iron-containing enzymes requires further study. Cancer cells depend on iron and ISC biogenesis, making ISC modulation a promising strategy for inducing ferroptosis. The inhibition of ISC synthesis leads to iron starvation, increasing ferroptotic cell death. Modulating ISCs through redox operations, iron chelation, or replacing iron with redox-inert metals provides potential therapeutic strategies for targeting cancer initiation and progression.

The pharmacological modulation of ISCs to induce ferroptosis is an area of promising research. However, several ISCs have been described as “repairable”, and the impact on normal cells should be considered when developing targeted drugs for ISCs. Combining ISC-targeted therapies with existing treatments, such as chemotherapy, radiotherapy, or immunotherapy, may enhance existing therapies and mitigate drug resistance. Further research is needed to develop and validate these strategies in clinical trials and determine their exact processes and efficacies in various cancers. Overall, targeting ISCs to induce ferroptosis is a promising cancer treatment strategy. Understanding the exact mechanisms and outcomes of this strategy will be an area of ongoing research and development.

## Figures and Tables

**Figure 1 cancers-15-02694-f001:**
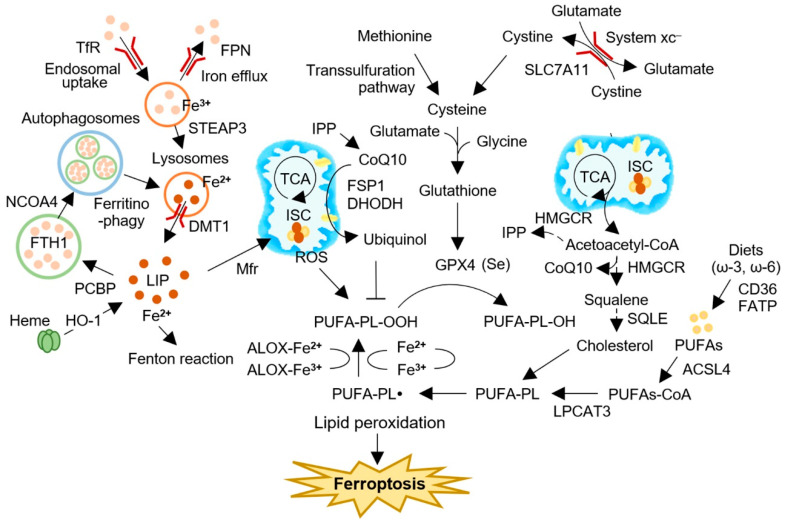
Iron metabolism and ferroptosis in cancer. Ferroptosis is a regulated form of cell death that is characterized by the buildup of lipid peroxides. The accumulation of labile iron and the subsequent generation of reactive oxygen species (ROS) via the Fenton reaction lead to oxidative damage of membrane lipids, particularly PUFA, resulting in lipid peroxidation and, ultimately, ferroptotic cell death. The canonical regulators of ferroptosis include the xCT, GPX4, FSP1-CoQ_10_, and DHODH, which have primary antioxidant functions and can detoxify cellular lipid peroxidation. GPX4 and CoQ_10_ synthesis, as well as other surveillance mechanisms that regulate lipid peroxidation, can be modulated by cholesterol synthesis and membrane remodeling. Iron metabolism and ferritinophagy, a type of autophagy mediated by iron, can also influence the intracellular pool of labile iron, which contains the most chelatable redox-active ferrous iron (Fe^2+^), leading to the generation of soluble radicals via the Fenton reaction. ACSL4, acyl-CoA synthetase long chain family member 4; ALOX, arachidonate lipoxygenase; BH4, tetrahydrobiopterin; CoQ_10_, coenzyme Q10 (ubiquinone-10); DHODH, dihydroorotate dehydrogenase; DMT1, divalent metal (iron) transporter 1; FATP, fatty acid transport protein; FSP1, ferroptosis suppressor protein 1; FTH1, ferritin heavy chain 1; GPX4, glutathione peroxidase 4; HMGCR, 3-hydroxy-3-methylglutaryl-CoA reductase; HO·, hydroxyl radical; HO-1, heme oxygenase 1; ISC, iron-sulfur cluster; IPP, isopentenyl pyrophosphate; ISC, iron-sulfur cluster; LIP, labile iron pool; Mfr, mitoferrin; NCOA4, nuclear receptor coactivator 4; PCBP, poly(C)-binding protein; PUFA-PL, polyunsaturated fatty acid-containing phospholipid; PL-PUFA-OH, polyunsaturated fatty acid-containing phospholipid alcohol; PUFAs, polyunsaturated fatty acids; ROS, reactive oxygen species; SQLE, squalene monooxygenase; SREBP1, sterol regulatory element-binding protein-1; TCA, tricarboxylic acid cycle; STEAP3, six-transmembrane epithelial antigen of prostate 3; TfR, transferrin receptor; xCT, system xc^–^ cystine/glutamate antiporter.

**Figure 2 cancers-15-02694-f002:**
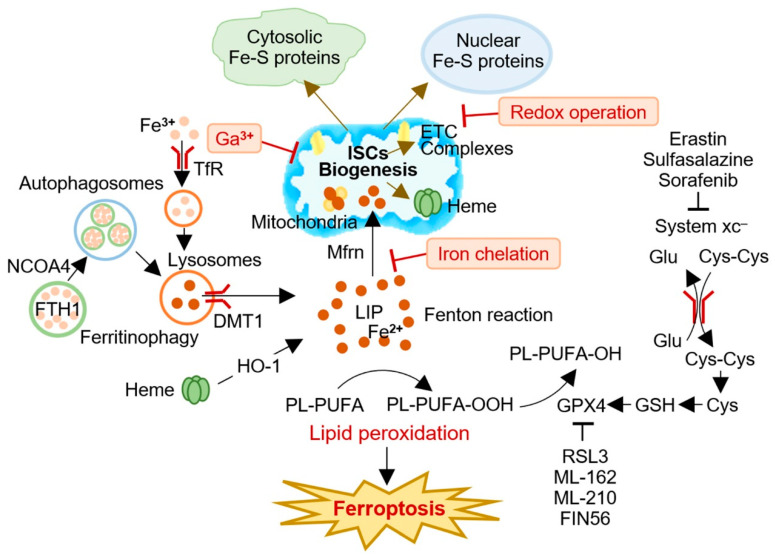
A conceptual illustration of targeting iron-sulfur clusters (ISCs) to induce ferroptosis in cancer cells. Cancer cells have a high demand for ISCs and iron, making modulation of ISC biogenesis an attractive strategy for ferroptosis-based cancer therapy. Several approaches, including redox operations, iron chelation, and iron replacement with redox-inert metals, have been proposed to impair the biogenesis and function of ISCs and induce ferroptosis in cancer cells. Targeting ISCs for ferroptosis induction represents a promising therapeutic avenue for cancer treatment. Cys, cysteine; Cys-Cys, cystine; DMT1 (SLC11A2), divalent metal (iron) transporter 1; ETC, electron transport chain; FPN (SLC40A1), ferroportin; Ga^3+^, gallium; Glu, glutamate; GPX4, glutathione peroxidase 4; GSH, glutathione; HO-1, heme oxygenase 1; ISC, iron-sulfur cluster; LIP, labile iron pool; Mfr, mitoferrin; NCOA4, nuclear receptor coactivator 4; PL-PUFA, polyunsaturated fatty acid-containing phospholipid; TfR, transferrin receptor; xCT, system xc^–^ cystine/glutamate antiporter.

## Data Availability

The data can be shared up on request.

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
