# Peer review of "Targeting Iron-Sulfur Clusters in Cancer: Opportunities and Challenges for Ferroptosis-Based Therapy"

_cancers, 2023, doi:10.3390/cancers15102694_

Round 1

Reviewer 1 Report

The review  focuses on how FeS cluster containing proteins can be involed into cancer treatment. Ferroptosis is a recently discovered mechanism of cell death: the review highlights how iron regulation is involved in lipid peroxidation and cancer proliferation and discuss the different Iron Sulfur Cluster containing proteins that are involved into iron and ROS accumulation, therefore triggering ferroptosis.  The last part of the review focuses on some of the strategies proposed up to date to target ISC proteins involved into ferroptosis-based cancer therapy.

Overall, the review is exahustive, very well organized and well written. The paper reanalyze several well characterized systems, such as the ISC machinery and the very many mitochondrial FeS proteins involved in these processes, on the perspective of this newly described cell death process. The article deserves publication provided a few correction/modifications here below summarized:

-Gallium is not a lanthanide metal. Correct the statement at bottom of page 9 (lines 418).

-There are some recent articles that have not been quoted and authors may wish to consider, such as

Li, Y., Wang, X., Huang, Z. et al. CISD3 inhibition drives cystine-deprivation induced ferroptosis. Cell Death Dis 12, 839 (2021). https://doi.org/10.1038/s41419-021-04128-2

 Jiang H, Fang Y, Wang Y, Li T, Lin H, Lin J, Pan T, Liu Q, Lv J, Chen D, Chen Y. FGF4 improves hepatocytes ferroptosis in autoimmune hepatitis mice via activation of CISD3. Int Immunopharmacol. 2023 Mar;116:109762. doi: 10.1016/j.intimp.2023.109762. Epub 2023 Jan 24. PMID: 36702076.

Furthermore, it would be appropriate to mention, even briefly, the very many spectroscopic studies in solution aiming at the characterization of cluster stability and cluster transfer in ISC containing proteins (for example, these have been reviewed in:  Banci L, Camponeschi F, Ciofi-Baffoni S, Piccioli M. The NMR contribution to protein-protein networking in Fe-S protein maturation. J Biol Inorg Chem. 2018 Jun;23(4):665-685. doi: 10.1007/s00775-018-1552-x. Epub 2018 Mar 22.

-The last section dealing with targeting ISC for ferroptosis-based cancer therapy focuses on the use of iron chelators and antioxidant molecules such as Vitamin C as useful strategies for treating human cancers. This reviewer is, admittedly, not an expert in this filed; however I feel that a broader overview could be appropriate in this section in order to cover more exhaustively the complex issue of LIP species.

Author Response

For Reviewer #1

The review  focuses on how FeS cluster containing proteins can be involed into cancer treatment. Ferroptosis is a recently discovered mechanism of cell death: the review highlights how iron regulation is involved in lipid peroxidation and cancer proliferation and discuss the different Iron Sulfur Cluster containing proteins that are involved into iron and ROS accumulation, therefore triggering ferroptosis.  The last part of the review focuses on some of the strategies proposed up to date to target ISC proteins involved into ferroptosis-based cancer therapy.

Overall, the review is exahustive, very well organized and well written. The paper reanalyze several well characterized systems, such as the ISC machinery and the very many mitochondrial FeS proteins involved in these processes, on the perspective of this newly described cell death process. The article deserves publication provided a few correction/modifications here below summarized:

-Gallium is not a lanthanide metal. Correct the statement at bottom of page 9 (lines 418).

Response: The word ‘lanthanide’ was deleted (line 443).

-There are some recent articles that have not been quoted and authors may wish to consider, such as

Li, Y., Wang, X., Huang, Z. et al. CISD3 inhibition drives cystine-deprivation induced ferroptosis. Cell Death Dis 12, 839 (2021). https://doi.org/10.1038/s41419-021-04128-2

 Jiang H, Fang Y, Wang Y, Li T, Lin H, Lin J, Pan T, Liu Q, Lv J, Chen D, Chen Y. FGF4 improves hepatocytes ferroptosis in autoimmune hepatitis mice via activation of CISD3. Int Immunopharmacol. 2023 Mar;116:109762. doi: 10.1016/j.intimp.2023.109762. Epub 2023 Jan 24. PMID: 36702076.

Response: As recommended, the CISD3 and related references were added in the revised text (page 6) and Refs #62-63.

Furthermore, it would be appropriate to mention, even briefly, the very many spectroscopic studies in solution aiming at the characterization of cluster stability and cluster transfer in ISC containing proteins (for example, these have been reviewed in:  Banci L, Camponeschi F, Ciofi-Baffoni S, Piccioli M. The NMR contribution to protein-protein networking in Fe-S protein maturation. J Biol Inorg Chem. 2018 Jun;23(4):665-685. doi: 10.1007/s00775-018-1552-x. Epub 2018 Mar 22.

Response: As noted, the spectroscopic studies were briefly mentioned, and the reference was added as #50 (page 5). Other references were correctly renumbered in the revised text and references.

-The last section dealing with targeting ISC for ferroptosis-based cancer therapy focuses on the use of iron chelators and antioxidant molecules such as Vitamin C as useful strategies for treating human cancers. This reviewer is, admittedly, not an expert in this filed; however I feel that a broader overview could be appropriate in this section in order to cover more exhaustively the complex issue of LIP species.

Response: We appreciate your kind comments. The last section has discussed the pharmacological modulation of ISCs to induce ferroptosis as an area of promising research. Modulating ISC through redox operations, iron chelation, or replacing iron with redox-inert metals provides potential therapeutic strategies for targeting human cancers. Other previous articles have extensively reviewed the complex issue of LIP species and the class IV ferroptosis-inducing compounds (FINs). Therefore, we have focused on targeting the ISCs for cancer therapy.

Reviewer 2 Report

The review provides insight into new therapeutic strategies for cancer treatment based on regulating ISC metabolism. The entry point is novel, and the content is substantial, which has certain reference significance for related theories and clinical research. There are some points to answer:

1. The introduction of ISC from line 76 to 77 can be put in the next paragraph from line 87.

2. The two sentences in lines 78 to 80 are repeated.

3. The normal function of iron is repeated from line 104 to 106.

4. In logical order, the paragraph that begins at line 116 can be arranged after the paragraph that begins at line 128.

5.  How does the paragraph from line 152 relate to tumors, ISC and ferroptosis? It should clarify the meaning of the subheading.

6. Why are NCOA4 and ferroptosis reintroduced in lines 158 to 159?

7. The description of the normal functions of ISC in lines 197 through 200 can be put together in the paragraph from line 87.

Author Response

For Reviewer #2

The review provides insight into new therapeutic strategies for cancer treatment based on regulating ISC metabolism. The entry point is novel, and the content is substantial, which has certain reference significance for related theories and clinical research. There are some points to answer:

  1. The introduction of ISC from line 76 to 77 can be put in the next paragraph from line 87.

Response: As recommended, the lines were moved to line 87. The references were renumbered.

  1. The two sentences in lines 78 to 80 are repeated.

Response: The lines 79-80 were deleted.

  1. The normal function of iron is repeated from line 104 to 106.

Response: Line 108 was deleted.

  1. In logical order, the paragraph that begins at line 116 can be arranged after the paragraph that begins at line 128.

Response: As recommended, the paragraphs were switched, and the references were correctly renumbered.

  1.  How does the paragraph from line 152 relate to tumors, ISC and ferroptosis? It should clarify the meaning of the subheading.

Response: Section 2 (iron metabolism and ferroptosis in cancer) describes iron and ferroptosis. The ISC and ferroptosis in cancer are well discussed in section 3 (ISC modulation and ferroptosis in cancer).

  1. Why are NCOA4 and ferroptosis reintroduced in lines 158 to 159?

Response: The NCOA4 is critical in ferroptosis initiation. This sentence would be better to remain with minor changes.

  1. The description of the normal functions of ISC in lines 197 through 200 can be put together in the paragraph from line 87.

Response: As noted, lines 197 through 200 combined the paragraphs from line 87. The references were renumbered.

Reviewer 3 Report

This manuscript presents a comprehensive review of iron metabolism, ferroptosis, and the potential of targeting iron-sulfur clusters (ISCs) for cancer therapy. The authors explain how the dysregulation of iron homeostasis is a hallmark of cancer, leading to excess reactive labile iron that can induce ferroptosis, a regulated form of cell death. They also suggest that cancer cells depend on iron and ISC biogenesis, making ISC modulation a promising strategy for inducing ferroptosis. It also provides potential therapeutic strategies for targeting cancer initiation and progression.

The strength of the article lies in its comprehensive overview of the role of iron in cancer and the potential of targeting ISCs for therapy. It discusses the mechanisms of ferroptosis and how it can be induced in cancer cells through ISC modulation. Additionally, the article provides potential therapeutic strategies for targeting cancer initiation and progression.

One weakness of the article is that it does not provide sufficient information about current or ongoing clinical trials. The authors suggest that targeting ISCs to induce ferroptosis is a promising cancer treatment strategy, but it would be more ideal if the authors included information about any current or ongoing studies performed in preclinical settings (in vivo models) or clinical trials to determine the efficacy of this approach in various cancers.

Author Response

Reviewer #3

This manuscript presents a comprehensive review of iron metabolism, ferroptosis, and the potential of targeting iron-sulfur clusters (ISCs) for cancer therapy. The authors explain how the dysregulation of iron homeostasis is a hallmark of cancer, leading to excess reactive labile iron that can induce ferroptosis, a regulated form of cell death. They also suggest that cancer cells depend on iron and ISC biogenesis, making ISC modulation a promising strategy for inducing ferroptosis. It also provides potential therapeutic strategies for targeting cancer initiation and progression.

The strength of the article lies in its comprehensive overview of the role of iron in cancer and the potential of targeting ISCs for therapy. It discusses the mechanisms of ferroptosis and how it can be induced in cancer cells through ISC modulation. Additionally, the article provides potential therapeutic strategies for targeting cancer initiation and progression.

One weakness of the article is that it does not provide sufficient information about current or ongoing clinical trials. The authors suggest that targeting ISCs to induce ferroptosis is a promising cancer treatment strategy, but it would be more ideal if the authors included information about any current or ongoing studies performed in preclinical settings (in vivo models) or clinical trials to determine the efficacy of this approach in various cancers.

Response: We appreciate your valuable comments. Pharmaceutical trials for modulating ISCs through redox operations, iron chelation, or replacing iron with redox-inert metals were reviewed in their preclinical and clinical settings in section 4 (pages 9-10). Most studies on ISC inhibitors have been involved only in the preclinical stage but have not reached clinical trials. Further validation of these efficacies for ferroptosis-based cancer therapy is needed in clinical trials. This was mentioned in the last paragraph of the revised section 4.